# Leveraged volume sampling for linear regression

**Michał Dereziński** and **Manfred K. Warmuth**
Department of Computer Science
University of California, Santa Cruz
mderezin@berkeley.edu, manfred@ucsc.edu

**Daniel Hsu**
Computer Science Department
Columbia University, New York
djhsu@cs.columbia.edu

## Abstract

Suppose an $n \times d$ design matrix in a linear regression problem is given, but the response for each point is hidden unless explicitly requested. The goal is to sample only a small number $k \ll n$ of the responses, and then produce a weight vector whose sum of squares loss over *all* points is at most $1 + \epsilon$ times the minimum. When $k$ is very small (e.g., $k = d$), jointly sampling diverse subsets of points is crucial. One such method called *volume sampling* has a unique and desirable property that the weight vector it produces is an unbiased estimate of the optimum. It is therefore natural to ask if this method offers the optimal unbiased estimate in terms of the number of responses $k$ needed to achieve a $1 + \epsilon$ loss approximation.

Surprisingly we show that volume sampling can have poor behavior when we require a very accurate approximation – indeed worse than some i.i.d. sampling techniques whose estimates are biased, such as *leverage score sampling*. We then develop a new rescaled variant of volume sampling that produces an unbiased estimate which avoids this bad behavior and has at least as good a tail bound as leverage score sampling: sample size $k = O(d \log d + d/\epsilon)$ suffices to guarantee total loss at most $1 + \epsilon$ times the minimum with high probability. Thus we improve on the best previously known sample size for an unbiased estimator, $k = O(d^2/\epsilon)$.

Our rescaling procedure leads to a new efficient algorithm for volume sampling which is based on a *determinantal rejection sampling* technique with potentially broader applications to determinantal point processes. Other contributions include introducing the combinatorics needed for rescaled volume sampling and developing tail bounds for sums of dependent random matrices which arise in the process.

## 1 Introduction

Consider a linear regression problem where the input points in $\mathbb{R}^d$ are provided, but the associated response for each point is withheld unless explicitly requested. The goal is to sample the responses for just a small subset of inputs, and then produce a weight vector whose total square loss on all $n$ points is at most $1 + \epsilon$ times that of the optimum.[1] This scenario is relevant in many applications where data points are cheap to obtain but responses are expensive. Surprisingly, with the aid of having all input points available, such multiplicative loss bounds are achievable without any range dependence on the points or responses common in on-line learning [see, e.g., 8].

A natural and intuitive approach to this problem is *volume sampling*, since it prefers "diverse" sets of points that will likely result in a weight vector with low total loss, regardless of what the corresponding responses turn out to be [11]. Volume sampling is closely related to optimal design criteria [18, 26], which are appropriate under statistical models of the responses; here we study a worst-case setting where the algorithm must use randomization to guard itself against worst-case responses.

Volume sampling and related determinantal point processes are employed in many machine learning and statistical contexts, including linear regression [11, 13, 26], clustering and matrix approximation [4, 14, 15], summarization and information retrieval [19, 23, 24], and fairness [6, 7]. The availability of fast algorithms for volume sampling [11, 26] has made it an important technique in the algorithmic toolbox alongside i.i.d. leverage score sampling [17] and spectral sparsification [5, 25].

It is therefore surprising that using volume sampling in the context of linear regression, as suggested in previous works [11, 26], may lead to suboptimal performance. We construct an example in which, even after sampling up to half of the responses, the loss of the weight vector from volume sampling is a fixed factor $>1$ larger than the minimum loss. Indeed, this poor behavior arises because for any sample size $>d$, the marginal probabilities from volume sampling are a mixture of uniform probabilities and leverage score probabilities, and uniform sampling is well-known to be suboptimal when the leverage scores are highly non-uniform.

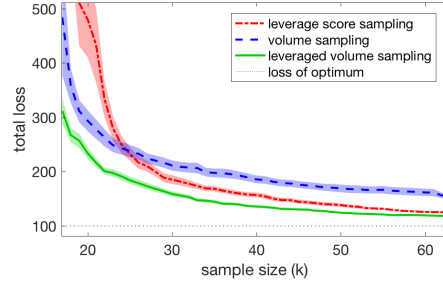

A possible recourse is to abandon volume sampling in favor of leverage score sampling [17, 33]. However, all i.i.d. sampling methods, including leverage score sampling, suffer from a coupon collector problem that prevents their effective use at small sample sizes [13]. Moreover, the resulting weight vectors are biased (when regarded as estimators for the least squares solution based on all responses). This is a nuisance when averaging multiple solutions (e.g., as produced in distributed settings). In contrast, volume sampling offers multiplicative loss bounds even with sample sizes as small as $d$ and it is the *only* known non-trivial method that gives unbiased weight vectors [11].

Figure 1: Plots of the total loss for the sampling methods (averaged over 100 runs) versus sample size (shading is standard error) for the libsvm dataset *cpusmall* [9].

We develop a new solution, called *leveraged volume sampling*, that retains the aforementioned benefits of volume sampling while avoiding its flaws. Specifically, we propose a variant of volume sampling based on rescaling the input points to "correct" the resulting marginals. On the algorithmic side, this leads to a new *determinantal rejection sampling* procedure which offers significant computational advantages over existing volume sampling algorithms, while at the same time being strikingly simple to implement. We prove that this new sampling scheme retains the benefits of volume sampling (like unbiasedness) but avoids the bad behavior demonstrated in our lower bound example. Along the way, we prove a new generalization of the Cauchy-Binet formula, which is needed for the rejection sampling denominator. Finally, we develop a new method for proving matrix tail bounds for leveraged volume sampling. Our analysis shows that the unbiased least-squares estimator constructed this way achieves a $1 + \epsilon$ approximation factor from a sample of size $O(d \log d + d/\epsilon)$, addressing an open question posed by [11].

**Experiments.**   Figure 1 presents experimental evidence on a benchmark dataset (*cpusmall* from the libsvm collection [9]) that the potential bad behavior of volume sampling proven in our lower bound does occur in practice. Appendix E shows more datasets and a detailed discussion of the experiments. In summary, leveraged volume sampling avoids the bad behavior of standard volume sampling, and performs considerably better than leverage score sampling, especially for small sample sizes $k$.

**Related work.**   Despite the ubiquity of volume sampling in many contexts already mentioned above, it has only recently been analyzed for linear regression. Focusing on small sample sizes, [11] proved multiplicative bounds for the expected loss of size $k = d$ volume sampling. Because the estimators produced by volume sampling are unbiased, averaging a number of such estimators produced an estimator based on a sample of size $k = O(d^2/\epsilon)$ with expected loss at most $1 + \epsilon$ times the optimum. It was shown in [13] that if the responses are assumed to be linear functions of the input points plus white noise, then size $k = O(d/\epsilon)$ volume sampling suffices for obtaining the same expected bounds. These noise assumptions on the response vector are also central to the task of A-optimal design, where volume sampling is a key technique [2, 18, 28, 29]. All of these previous results were concerned with bounds that hold in expectation; it is natural to ask if similar (or better) bounds can also be shown to hold with high probability, without noise assumptions. Concentration bounds for volume sampling and other strong Rayleigh measures were studied in [30], but these results are not sufficient to obtain the tail bounds for volume sampling.

Other techniques applicable to our linear regression problem include leverage score sampling [17] and spectral sparsification [5, 25]. Leverage score sampling is an i.i.d. sampling procedure which achieves tail bounds matching the ones we obtain here for leveraged volume sampling, however it produces biased weight vectors and experimental results (see [13] and Appendix E) show that it has weaker performance for small sample sizes. A different and more elaborate sampling technique based on spectral sparsification [5, 25] was recently shown to be effective for linear regression [10], however this method also does not produce unbiased estimates, which is a primary concern of this paper and desirable in many settings. Unbiasedness seems to require delicate control of the sampling probabilities, which we achieve using determinantal rejection sampling.

**Outline and contributions.** We set up our task of subsampling for linear regression in the next section and present our lower bound for standard volume sampling. A new variant of rescaled volume sampling is introduced in Section 3. We develop techniques for proving matrix expectation formulas for this variant which show that for any rescaling the weight vector produced for the subproblem is unbiased.

Next, we show that when rescaling with leverage scores, then a new algorithm based on rejection sampling is surprisingly efficient (Section 4): Other than the preprocessing step of computing leverage scores, the runtime does not depend on $n$ (a major improvement over existing volume sampling algorithms). Then, in Section 4.1 we prove multiplicative loss bounds for leveraged volume sampling by establishing two important properties which are hard to prove for joint sampling procedures. We conclude in Section 5 with an open problem and with a discussion of how rescaling with approximate leverage scores gives further time improvements for constructing an unbiased estimator.

## 2 Volume sampling for linear regression

In this section, we describe our linear regression setting, and review the guarantees that standard volume sampling offers in this context. Then, we present a surprising lower bound which shows that under worst-case data, this method can exhibit undesirable behavior.

### 2.1 Setting

Suppose the learner is given $n$ input vectors $\mathbf{x}_1, \ldots, \mathbf{x}_n \in \mathbb{R}^d$, which are arranged as the rows of an $n \times d$ input matrix $\mathbf{X}$. Each input vector $\mathbf{x}_i$ has an associated response variable $y_i \in \mathbb{R}$ from the response vector $\mathbf{y} \in \mathbb{R}^n$. The goal of the learner is to find a weight vector $\mathbf{w} \in \mathbb{R}^d$ that minimizes the square loss:

$$\mathbf{w}^* \overset{def}{=} \underset{\mathbf{w} \in \mathbb{R}^d}{\operatorname{argmin}} L(\mathbf{w}), \quad \text{where } L(\mathbf{w}) \overset{def}{=} \sum_{i=1}^{n} (\mathbf{x}_i^\top \mathbf{w} - y_i)^2 = \|\mathbf{X}\mathbf{w} - \mathbf{y}\|^2.$$

Given both matrix $\mathbf{X}$ and vector $\mathbf{y}$, the least squares solution can be directly computed as $\mathbf{w}^* = \mathbf{X}^+ \mathbf{y}$, where $\mathbf{X}^+$ is the pseudo-inverse. Throughout the paper we assume w.l.o.g. that $\mathbf{X}$ has (full) rank $d$.[2]

In our setting, the learner is initially given the entire input matrix $\mathbf{X}$, while response vector $\mathbf{y}$ remains hidden. The learner is then allowed to select a subset $S$ of row indices in $[n] = \{1, \ldots, n\}$ for which the corresponding responses $y_i$ are revealed. The learner next constructs an estimate $\widehat{\mathbf{w}}$ of $\mathbf{w}^*$ using matrix $\mathbf{X}$ and the partial vector of observed responses. Finally, the learner is evaluated by the loss over all rows of $\mathbf{X}$ (including the ones with unobserved responses), and the goal is to obtain a multiplicative loss bound, i.e., that for some $\epsilon > 0$,

$$L(\widehat{\mathbf{w}}) \leq (1 + \epsilon) L(\mathbf{w}^*).$$

### 2.2 Standard volume sampling

Given $\mathbf{X} \in \mathbb{R}^{n \times d}$ and a size $k \geq d$, standard volume sampling jointly chooses a set $S$ of $k$ indices in $[n]$ with probability

$$\Pr(S) = \frac{\det(\mathbf{X}_S^\top \mathbf{X}_S)}{\binom{n-d}{k-d} \det(\mathbf{X}^\top \mathbf{X})},$$

where $\mathbf{X}_S$ is the submatrix of the rows from $\mathbf{X}$ indexed by the set $S$. The learner then obtains the responses $y_i$, for $i \in S$, and uses the optimum solution $\mathbf{w}_S^* = (\mathbf{X}_S)^+ \mathbf{y}_S$ for the subproblem $(\mathbf{X}_S, \mathbf{y}_S)$ as its weight vector. The sampling procedure can be performed using *reverse iterative sampling* (shown on the right), which, if carefully implemented, takes $O(nd^2)$ time (see [11, 13]).

The key property (unique to volume sampling) is that the subsampled estimator $\mathbf{w}_S^*$ is unbiased, i.e.

$$\mathbb{E}[\mathbf{w}_S^*] = \mathbf{w}^*, \quad \text{where} \quad \mathbf{w}^* = \operatorname*{argmin}_{\mathbf{w}} L(\mathbf{w}).$$

As discussed in [11], this property has important practical implications in distributed settings: Mixtures of unbiased estimators remain unbiased (and can conveniently be used to reduce variance). Also if the rows of $\mathbf{X}$ are in general position, then for volume sampling

$$\mathbb{E}\big[(\mathbf{X}_S^\top \mathbf{X}_S)^{-1}\big] = \frac{n-d+1}{k-d+1}(\mathbf{X}^\top \mathbf{X})^{-1}. \tag{1}$$

| **Reverse iterative sampling** |
| --- |
| VolumeSample($\mathbf{X}$, $k$): $S \leftarrow [n]$ **while** $\lvert S \rvert > k$ $\quad \forall_{i \in S}: q_i \leftarrow \frac{\det(\mathbf{X}_{S \setminus i}^\top \mathbf{X}_{S \setminus i})}{\det(\mathbf{X}_S^\top \mathbf{X}_S)}$ $\quad$ Sample $i \propto q_i$ out of $S$ $\quad S \leftarrow S \setminus \{i\}$ **end** **return** $S$ |

This is important because in A-optimal design bounding $\operatorname{tr}((\mathbf{X}_S^\top \mathbf{X}_S)^{-1})$ is the main concern. Given these direct connections of volume sampling to linear regression, it is natural to ask whether this distribution achieves a loss bound of $(1 + \epsilon)$ times the optimum for small sample sizes $k$.

## 2.3 Lower bound for standard volume sampling

We show that standard volume sampling cannot guarantee $1 + \epsilon$ multiplicative loss bounds on some instances, unless over half of the rows are chosen to be in the subsample.

**Theorem 1** *Let $(\mathbf{X}, \mathbf{y})$ be an $n \times d$ least squares problem, such that*

$$\mathbf{X} = \begin{pmatrix} \mathbf{I}_{d \times d} \\ \hline \gamma\,\mathbf{I}_{d \times d} \\ \hline \vdots \\ \hline \gamma\,\mathbf{I}_{d \times d} \end{pmatrix}, \quad \mathbf{y} = \begin{pmatrix} \mathbf{1}_d \\ \hline \mathbf{0}_d \\ \hline \vdots \\ \hline \mathbf{0}_d \end{pmatrix}, \quad \text{where} \quad \gamma > 0.$$

*Let $\mathbf{w}_S^* = (\mathbf{X}_S)^+ \mathbf{y}_S$ be obtained from size $k$ volume sampling for $(\mathbf{X}, \mathbf{y})$. Then,*

$$\lim_{\gamma \to 0} \frac{\mathbb{E}[L(\mathbf{w}_S^*)]}{L(\mathbf{w}^*)} \geq 1 + \frac{n-k}{n-d}, \tag{2}$$

*and there is a $\gamma > 0$ such that for any $k \leq \frac{n}{2}$,*

$$\Pr\left(L(\mathbf{w}_S^*) \geq \left(1 + \frac{1}{2}\right) L(\mathbf{w}^*)\right) > \frac{1}{4}. \tag{3}$$

**Proof** In Appendix A we show (2), and that for the chosen $(\mathbf{X}, \mathbf{y})$ we have $L(\mathbf{w}^*) = \sum_{i=1}^d (1 - l_i)$ (see (8)), where $l_i = \mathbf{x}_i^\top (\mathbf{X}^\top \mathbf{X})^{-1} \mathbf{x}_i$ is the $i$-th leverage score of $\mathbf{X}$. Here, we show (3). The marginal probability of the $i$-th row under volume sampling (as given by [12]) is

$$\Pr(i \in S) = \theta\, l_i + (1 - \theta)\, 1 = 1 - \theta\,(1 - l_i), \quad \text{where } \theta = \frac{n-k}{n-d}. \tag{4}$$

Next, we bound the probability that all of the first $d$ input vectors were selected by volume sampling:

$$\Pr\big([d] \subseteq S\big) \overset{(*)}{\leq} \prod_{i=1}^d \Pr(i \in S) = \prod_{i=1}^d \left(1 - \frac{n-k}{n-d}(1 - l_i)\right) \leq \exp\left(-\frac{n-k}{n-d}\overbrace{\underbrace{L(\mathbf{w}^*)}}^{\sum_{i=1}^d (1-l_i)}\right),$$

where $(*)$ follows from negative associativity of volume sampling (see [26]). If for some $i \in [d]$ we have $i \notin S$, then $L(\mathbf{w}_S^*) \geq 1$. So for $\gamma$ such that $L(\mathbf{w}^*) = \frac{2}{3}$ and any $k \leq \frac{n}{2}$:

$$\Pr\left(L(\mathbf{w}_S^*) \geq \left(1 + \frac{1}{2}\right)\overbrace{L(\mathbf{w}^*)}^{2/3}\right) \geq 1 - \exp\left(-\frac{n-k}{n-d} \cdot \frac{2}{3}\right) \geq 1 - \exp\left(-\frac{1}{2} \cdot \frac{2}{3}\right) > \frac{1}{4}. \qquad \blacksquare$$

Note that this lower bound only makes use of the negative associativity of volume sampling and the form of the marginals. However the tail bounds we prove in Section 4.1 rely on more subtle properties of volume sampling. We begin by creating a variant of volume sampling with rescaled marginals.

## 3 Rescaled volume sampling

Given any size $k \geq d$, our goal is to jointly sample $k$ row indices $\pi_1, \ldots, \pi_k$ with replacement (instead of a *subset* $S$ of $[n]$ of size $k$, we get a *sequence* $\pi \in [n]^k$). The second difference to standard volume sampling is that we rescale the $i$-th row (and response) by $\frac{1}{\sqrt{q_i}}$, where $q = (q_1, \ldots, q_n)$ is any discrete distribution over the set of row indices $[n]$, such that $\sum_{i=1}^{n} q_i = 1$ and $q_i > 0$ for all $i \in [n]$. We now define $q$-rescaled size $k$ volume sampling as a joint sampling distribution over $\pi \in [n]^k$, s.t.

$$q\text{-rescaled size } k \text{ volume sampling:} \qquad \Pr(\pi) \sim \det\left(\sum_{i=1}^{k} \frac{1}{q_{\pi_i}} \mathbf{x}_{\pi_i} \mathbf{x}_{\pi_i}^\top\right) \prod_{i=1}^{k} q_{\pi_i}. \tag{5}$$

Using the following rescaling matrix $\mathbf{Q}_\pi \overset{def}{=} \sum_{i=1}^{|\pi|} \frac{1}{q_{\pi_i}} \mathbf{e}_{\pi_i} \mathbf{e}_{\pi_i}^\top \in \mathbb{R}^{n \times n}$, we rewrite the determinant as $\det(\mathbf{X}^\top \mathbf{Q}_\pi \mathbf{X})$. As in standard volume sampling, the normalization factor in rescaled volume sampling can be given in a closed form through a novel extension of the Cauchy-Binet formula (proof in Appendix B.1).

**Proposition 2** *For any $\mathbf{X} \in \mathbb{R}^{n \times d}$, $k \geq d$ and $q_1, \ldots, q_n > 0$, such that $\sum_{i=1}^{n} q_i = 1$, we have*

$$\sum_{\pi \in [n]^k} \det(\mathbf{X}^\top \mathbf{Q}_\pi \mathbf{X}) \prod_{i=1}^{k} q_{\pi_i} = k(k-1) \cdots (k-d+1) \det(\mathbf{X}^\top \mathbf{X}).$$

Given a matrix $\mathbf{X} \in \mathbb{R}^{n \times d}$, vector $\mathbf{y} \in \mathbb{R}^n$ and a sequence $\pi \in [n]^k$, we are interested in a least-squares problem $(\mathbf{Q}_\pi^{1/2} \mathbf{X}, \mathbf{Q}_\pi^{1/2} \mathbf{y})$, which selects instances indexed by $\pi$, and rescales each of them by the corresponding $1/\sqrt{q_i}$. This leads to a natural subsampled least squares estimator

$$\mathbf{w}_\pi^* = \operatorname*{argmin}_{\mathbf{w}} \sum_{i=1}^{k} \frac{1}{q_{\pi_i}} \left(\mathbf{x}_{\pi_i}^\top \mathbf{w} - y_{\pi_i}\right)^2 = (\mathbf{Q}_\pi^{1/2} \mathbf{X})^+ \mathbf{Q}_\pi^{1/2} \mathbf{y}.$$

The key property of standard volume sampling is that the subsampled least-squares estimator is unbiased. Surprisingly this property is retained for any $q$-rescaled volume sampling (proof in Section 3.1). As we shall see, this will give us great leeway for choosing $q$ to optimize our algorithms.

**Theorem 3** *Given a full rank $\mathbf{X} \in \mathbb{R}^{n \times d}$ and a response vector $\mathbf{y} \in \mathbb{R}^n$, for any $q$ as above, if $\pi$ is sampled according to (5), then*

$$\mathbb{E}[\mathbf{w}_\pi^*] = \mathbf{w}^*, \quad \text{where} \quad \mathbf{w}^* = \operatorname*{argmin}_{\mathbf{w}} \|\mathbf{X}\mathbf{w} - \mathbf{y}\|^2.$$

The matrix expectation equation (1) for standard volume sampling (discussed in Section 2) has a natural extension to any rescaled volume sampling, but now the equality turns into an inequality (proof in Appendix B.2):

**Theorem 4** *Given a full rank $\mathbf{X} \in \mathbb{R}^{n \times d}$ and any $q$ as above, if $\pi$ is sampled according to (5), then*

$$\mathbb{E}\left[(\mathbf{X}^\top \mathbf{Q}_\pi \mathbf{X})^{-1}\right] \preceq \frac{1}{k-d+1} (\mathbf{X}^\top \mathbf{X})^{-1}.$$

### 3.1 Proof of Theorem 3

We show that the least-squares estimator $\mathbf{w}_\pi^* = (\mathbf{Q}_\pi^{1/2} \mathbf{X})^+ \mathbf{Q}_\pi^{1/2} \mathbf{y}$ produced from any $q$-rescaled volume sampling is unbiased, illustrating a proof technique which is also useful for showing Theorem 4, as well as Propositions 2 and 5. The key idea is to apply the pseudo-inverse expectation formula for standard volume sampling (see e.g., [11]) first on the subsampled estimator $\mathbf{w}_\pi^*$, and then again on the full estimator $\mathbf{w}^*$. In the first step, this formula states:

$$\overbrace{(\mathbf{Q}_\pi^{1/2} \mathbf{X})^+ \mathbf{Q}_\pi^{1/2} \mathbf{y}}^{\mathbf{w}_\pi^*} = \sum_{S \in \binom{[k]}{d}} \frac{\det(\mathbf{X}^\top \mathbf{Q}_{\pi_S} \mathbf{X})}{\det(\mathbf{X}^\top \mathbf{Q}_\pi \mathbf{X})} \overbrace{(\mathbf{Q}_{\pi_S}^{1/2} \mathbf{X})^+ \mathbf{Q}_{\pi_S}^{1/2} \mathbf{y}}^{\mathbf{w}_{\pi_S}^*},$$

where $\binom{[k]}{d} \stackrel{def}{=} \{S \subseteq \{1,\ldots,k\} : |S| = d\}$ and $\pi_S$ denotes a subsequence of $\pi$ indexed by the elements of set $S$. Note that since $S$ is of size $d$, we can decompose the determinant:

$$\det(\mathbf{X}^\top \mathbf{Q}_{\pi_S} \mathbf{X}) = \det(\mathbf{X}_{\pi_S})^2 \prod_{i \in S} \frac{1}{q_{\pi_i}}.$$

Whenever this determinant is non-zero, $\mathbf{w}^*_{\pi_S}$ is the exact solution of a system of $d$ linear equations:

$$\frac{1}{\sqrt{q_{\pi_i}}} \mathbf{x}^\top_{\pi_i} \mathbf{w} = \frac{1}{\sqrt{q_{\pi_i}}} y_{\pi_i}, \qquad \text{for} \quad i \in S.$$

Thus, the rescaling of each equation by $\frac{1}{\sqrt{q_{\pi_i}}}$ cancels out, and we can simply write $\mathbf{w}^*_{\pi_S} = (\mathbf{X}_{\pi_S})^+ \mathbf{y}_{\pi_S}$. (Note that this is not the case for sets larger than $d$ whenever the optimum solution incurs positive loss.) We now proceed with summing over all $\pi \in [n]^k$. Following Proposition 2, we define the normalization constant as $Z = d!\binom{k}{d}\det(\mathbf{X}^\top \mathbf{X})$, and obtain:

$$Z\,\mathbb{E}[\mathbf{w}^*_\pi] = \sum_{\pi \in [n]^k} \left( \prod_{i=1}^k q_{\pi_i} \right) \det(\mathbf{X}^\top \mathbf{Q}_\pi \mathbf{X})\, \mathbf{w}^*_\pi = \sum_{\pi \in [n]^k} \sum_{S \in \binom{[k]}{d}} \left( \prod_{i \in [k]\setminus S} q_{\pi_i} \right) \det(\mathbf{X}_{\pi_S})^2 (\mathbf{X}_{\pi_S})^+ \mathbf{y}_{\pi_S}$$

$$\stackrel{(1)}{=} \binom{k}{d} \sum_{\bar{\pi} \in [n]^d} \det(\mathbf{X}_{\bar{\pi}})^2 (\mathbf{X}_{\bar{\pi}})^+ \mathbf{y}_{\bar{\pi}} \sum_{\tilde{\pi} \in [n]^{k-d}} \prod_{i=1}^{k-d} q_{\tilde{\pi}_i}$$

$$\stackrel{(2)}{=} \binom{k}{d} d! \sum_{S \in \binom{[n]}{d}} \det(\mathbf{X}_S)^2 (\mathbf{X}_S)^+ \mathbf{y}_S \left( \sum_{i=1}^n q_i \right)^{k-d} \stackrel{(3)}{=} \overbrace{\binom{k}{d} d!\det(\mathbf{X}^\top \mathbf{X})}^{Z}\, \mathbf{w}^*.$$

Note that in (1) we separate $\pi$ into two parts, $\bar{\pi}$ and $\tilde{\pi}$ (respectively, for subsets $S$ and $[k]\setminus S$), and sum over them separately. The binomial coefficient $\binom{k}{d}$ counts the number of ways that $S$ can be "placed into" the sequence $\pi$. In (2) we observe that whenever $\bar{\pi}$ has repetitions, determinant $\det(\mathbf{X}_{\bar{\pi}})$ is zero, so we can switch to summing over sets. Finally, (3) again uses the standard size $d$ volume sampling unbiasedness formula, now for the least-squares task $(\mathbf{X}, \mathbf{y})$, and the fact that $q_i$'s sum to 1.

## 4 Leveraged volume sampling: a natural rescaling

Rescaled volume sampling can be viewed as selecting a sequence $\pi$ of $k$ rank-1 matrices from the covariance matrix $\mathbf{X}^\top \mathbf{X} = \sum_{i=1}^n \mathbf{x}_i \mathbf{x}_i^\top$. If $\pi_1,\ldots,\pi_k$ are sampled i.i.d. from $q$, i.e., $\Pr(\pi) = \prod_{i=1}^k q_{\pi_i}$, then matrix $\frac{1}{k}\mathbf{X}^\top \mathbf{Q}_\pi \mathbf{X}$ is an unbiased estimator of the covariance matrix because $\mathbb{E}[q_{\pi_i}^{-1}\mathbf{x}_{\pi_i}\mathbf{x}_{\pi_i}^\top] = \mathbf{X}^\top \mathbf{X}$. In rescaled volume sampling (5), $\Pr(\pi) \sim \left(\prod_{i=1}^k q_{\pi_i}\right)\frac{\det(\mathbf{X}^\top \mathbf{Q}_\pi \mathbf{X})}{\det(\mathbf{X}^\top \mathbf{X})}$, and the latter volume ratio introduces a bias to that estimator. However, we show

---

**Determinantal rejection sampling**

1: **Input:** $\mathbf{X} \in \mathbb{R}^{n \times d}, q = (\frac{l_1}{d},\ldots,\frac{l_n}{d}), k \geq d$
2: $s \leftarrow \max\{k,\,4d^2\}$
3: **repeat**
4:     Sample $\pi_1,\ldots,\pi_s$ i.i.d. $\sim (q_1,\ldots,q_n)$
5:     Sample $Accept \sim$ Bernoulli$\left(\frac{\det(\frac{1}{s}\mathbf{X}^\top \mathbf{Q}_\pi \mathbf{X})}{\det(\mathbf{X}^\top \mathbf{X})}\right)$
6: **until** $Accept =$ true
7: $S \leftarrow$ VolumeSample$\left((\mathbf{Q}^{1/2}_{[1..n]}\mathbf{X})_\pi, k\right)$
8: **return** $\pi_S$

---

that this bias vanishes when $q$ is exactly proportional to the leverage scores (proof in Appendix B.3).

**Proposition 5** *For any $q$ and $\mathbf{X}$ as before, if $\pi \in [n]^k$ is sampled according to (5), then*

$$\mathbb{E}[\mathbf{Q}_\pi] = (k-d)\,\mathbf{I} + \mathrm{diag}\left(\frac{l_1}{q_1},\ldots,\frac{l_n}{q_n}\right), \quad \text{where} \quad l_i \stackrel{def}{=} \mathbf{x}_i^\top (\mathbf{X}^\top \mathbf{X})^{-1}\mathbf{x}_i.$$

*In particular, $\mathbb{E}[\frac{1}{k}\mathbf{X}^\top \mathbf{Q}_\pi \mathbf{X}] = \mathbf{X}^\top \mathbb{E}[\frac{1}{k}\mathbf{Q}_\pi]\mathbf{X} = \mathbf{X}^\top \mathbf{X}$ if and only if $q_i = \frac{l_i}{d} > 0$ for all $i \in [n]$.*

This special rescaling, which we call *leveraged volume sampling*, has other remarkable properties. Most importantly, it leads to a simple and efficient algorithm we call *determinantal rejection sampling*: Repeatedly sample $O(d^2)$ indices $\pi_1,\ldots,\pi_s$ i.i.d. from $q = (\frac{l_1}{d},\ldots,\frac{l_n}{d})$, and accept the sample with probability proportional to its volume ratio. Having obtained a sample, we can further reduce its size via reverse iterative sampling. We show next that this procedure not only returns a $q$-rescaled volume sample, but also exploiting the fact that $q$ is proportional to the leverage scores, it requires (surprisingly) only a constant number of iterations of rejection sampling with high probability.

**Theorem 6** *Given the leverage score distribution $q = (\frac{l_1}{d}, \ldots, \frac{l_n}{d})$ and the determinant $\det(\mathbf{X}^\top \mathbf{X})$ for matrix $\mathbf{X} \in \mathbb{R}^{n \times d}$, determinantal rejection sampling returns sequence $\pi_S$ distributed according to leveraged volume sampling, and w.p. at least $1 - \delta$ finishes in time $O((d^2 + k)d^2 \ln(\frac{1}{\delta}))$.*

**Proof** We use a composition property of rescaled volume sampling (proof in Appendix B.4):

**Lemma 7** *Consider the following sampling procedure, for $s > k$:*

$$\pi \overset{s}{\sim} \mathbf{X} \qquad\qquad\qquad \textit{(q-rescaled size s volume sampling)},$$

$$S \overset{k}{\sim} \begin{pmatrix} \frac{1}{\sqrt{q_{\pi_1}}} \mathbf{x}_{\pi_1}^\top \\ \cdots \\ \frac{1}{\sqrt{q_{\pi_s}}} \mathbf{x}_{\pi_s}^\top \end{pmatrix} = \left( \mathbf{Q}_{[1..n]}^{1/2} \mathbf{X} \right)_\pi \qquad \textit{(standard size k volume sampling)}.$$

*Then $\pi_S$ is distributed according to $q$-rescaled size $k$ volume sampling from $\mathbf{X}$.*

First, we show that the rejection sampling probability in line 5 of the algorithm is bounded by 1:

$$\frac{\det(\frac{1}{s}\mathbf{X}^\top \mathbf{Q}_\pi \mathbf{X})}{\det(\mathbf{X}^\top \mathbf{X})} = \det\left( \frac{1}{s}\mathbf{X}^\top \mathbf{Q}_\pi \mathbf{X}(\mathbf{X}^\top \mathbf{X})^{-1} \right) \overset{(*)}{\leq} \left( \frac{1}{d}\text{tr}\left( \frac{1}{s}\mathbf{X}^\top \mathbf{Q}_\pi \mathbf{X}(\mathbf{X}^\top \mathbf{X})^{-1} \right) \right)^d$$

$$= \left( \frac{1}{ds}\text{tr}\left( \mathbf{Q}_\pi \mathbf{X}(\mathbf{X}^\top \mathbf{X})^{-1}\mathbf{X}^\top \right) \right)^d = \left( \frac{1}{ds}\sum_{i=1}^s \frac{d}{l_i}\mathbf{x}_i^\top(\mathbf{X}^\top \mathbf{X})^{-1}\mathbf{x}_i \right)^d = 1,$$

where $(*)$ follows from the geometric-arithmetic mean inequality for the eigenvalues of the underlying matrix. This shows that sequence $\pi$ is drawn according to $q$-rescaled volume sampling of size $s$. Now, Lemma 7 implies correctness of the algorithm. Next, we use Proposition 2 to compute the expected value of acceptance probability from line 5 under the i.i.d. sampling of line 4:

$$\sum_{\pi \in [n]^s} \left( \prod_{i=1}^s q_{\pi_i} \right) \frac{\det(\frac{1}{s}\mathbf{X}^\top \mathbf{Q}_\pi \mathbf{X})}{\det(\mathbf{X}^\top \mathbf{X})} = \frac{s(s-1)\ldots(s-d+1)}{s^d} \geq \left( 1 - \frac{d}{s} \right)^d \geq 1 - \frac{d^2}{s} \geq \frac{3}{4},$$

where we also used Bernoulli's inequality and the fact that $s \geq 4d^2$ (see line 2). Since the expected value of the acceptance probability is at least $\frac{3}{4}$, an easy application of Markov's inequality shows that at each trial there is at least a 50% chance of it being above $\frac{1}{2}$. So, the probability of at least $r$ trials occurring is less than $(1 - \frac{1}{4})^r$. Note that the computational cost of one trial is no more than the cost of SVD decomposition of matrix $\mathbf{X}^\top \mathbf{Q}_\pi \mathbf{X}$ (for computing the determinant), which is $O(sd^2)$. The cost of reverse iterative sampling (line 7) is also $O(sd^2)$ with high probability (as shown by [13]). Thus, the overall runtime is $O((d^2 + k)d^2 r)$, where $r \leq \ln(\frac{1}{\delta})/\ln(\frac{4}{3})$ w.p. at least $1 - \delta$. ∎

### 4.1 Tail bounds for leveraged volume sampling

An analysis of leverage score sampling, essentially following [33, Section 2] which in turn draws from [31], highlights two basic sufficient conditions on the (random) subsampling matrix $\mathbf{Q}_\pi$ that lead to multiplicative tail bounds for $L(\mathbf{w}_\pi^*)$.

It is convenient to shift to an orthogonalization of the linear regression task $(\mathbf{X}, \mathbf{y})$ by replacing matrix $\mathbf{X}$ with a matrix $\mathbf{U} = \mathbf{X}(\mathbf{X}^\top \mathbf{X})^{-1/2} \in \mathbb{R}^{n \times d}$. It is easy to check that the columns of $\mathbf{U}$ have unit length and are orthogonal, i.e., $\mathbf{U}^\top \mathbf{U} = \mathbf{I}$. Now, $\mathbf{v}^* = \mathbf{U}^\top \mathbf{y}$ is the least-squares solution for the orthogonal problem $(\mathbf{U}, \mathbf{y})$ and prediction vector $\mathbf{U}\mathbf{v}^* = \mathbf{U}\mathbf{U}^\top \mathbf{y}$ for $(\mathbf{U}, \mathbf{y})$ is the same as the prediction vector $\mathbf{X}\mathbf{w}^* = \mathbf{X}(\mathbf{X}^\top \mathbf{X})^{-1}\mathbf{X}^\top \mathbf{y}$ for the original problem $(\mathbf{X}, \mathbf{y})$. The same property holds for the subsampled estimators, i.e., $\mathbf{U}\mathbf{v}_\pi^* = \mathbf{X}\mathbf{w}_\pi^*$, where $\mathbf{v}_\pi^* = (\mathbf{Q}_\pi^{1/2}\mathbf{U})^+ \mathbf{Q}_\pi^{1/2}\mathbf{y}$. Volume sampling probabilities are also preserved under this transformation, so w.l.o.g. we can work with the orthogonal problem. Now $L(\mathbf{v}_\pi^*)$ can be rewritten as

$$L(\mathbf{v}_\pi^*) = \|\mathbf{U}\mathbf{v}_\pi^* - \mathbf{y}\|^2 \overset{(1)}{=} \|\mathbf{U}\mathbf{v}^* - \mathbf{y}\|^2 + \|\mathbf{U}(\mathbf{v}_\pi^* - \mathbf{v}^*)\|^2 \overset{(2)}{=} L(\mathbf{v}^*) + \|\mathbf{v}_\pi^* - \mathbf{v}^*\|^2, \quad (6)$$

where (1) follows via Pythagorean theorem from the fact that $\mathbf{U}(\mathbf{v}_\pi^* - \mathbf{v}^*)$ lies in the column span of $\mathbf{U}$ and the residual vector $\mathbf{r} = \mathbf{U}\mathbf{v}^* - \mathbf{y}$ is orthogonal to all columns of $\mathbf{U}$, and (2) follows from $\mathbf{U}^\top \mathbf{U} = \mathbf{I}$. By the definition of $\mathbf{v}_\pi^*$, we can write $\|\mathbf{v}_\pi^* - \mathbf{v}^*\|$ as follows:

$$\|\mathbf{v}_\pi^* - \mathbf{v}^*\| = \|(\mathbf{U}^\top \mathbf{Q}_\pi \mathbf{U})^{-1} \underset{d \times d}{\mathbf{U}^\top \mathbf{Q}_\pi(\mathbf{y} - \mathbf{U}\mathbf{v}^*)}\| \leq \|(\mathbf{U}^\top \mathbf{Q}_\pi \mathbf{U})^{-1}\| \underset{d \times 1}{\|\mathbf{U}^\top \mathbf{Q}_\pi \mathbf{r}\|}, \quad (7)$$

where $\|\mathbf{A}\|$ denotes the matrix 2-norm (i.e., the largest singular value) of $\mathbf{A}$; when $\mathbf{A}$ is a vector, then $\|\mathbf{A}\|$ is its Euclidean norm. This breaks our task down to showing two key properties:

1. *Matrix multiplication:* Upper bounding the Euclidean norm $\|\mathbf{U}^\top \mathbf{Q}_\pi \mathbf{r}\|$,
2. *Subspace embedding:* Upper bounding the matrix 2-norm $\|(\mathbf{U}^\top \mathbf{Q}_\pi \mathbf{U})^{-1}\|$.

We start with a theorem that implies strong guarantees for approximate matrix multiplication with leveraged volume sampling. Unlike with i.i.d. sampling, this result requires controlling the pairwise dependence between indices selected under rescaled volume sampling. Its proof is an interesting application of a classical Hadamard matrix product inequality from [3] (Proof in Appendix C).

**Theorem 8** *Let* $\mathbf{U} \in \mathbb{R}^{n \times d}$ *be a matrix s.t.* $\mathbf{U}^\top \mathbf{U} = \mathbf{I}$*. If sequence* $\pi \in [n]^k$ *is selected using leveraged volume sampling of size* $k \geq \frac{2d}{\epsilon}$*, then for any* $\mathbf{r} \in \mathbb{R}^n$*,*

$$\mathbb{E}\left[\left\|\frac{1}{k}\mathbf{U}^\top \mathbf{Q}_\pi \mathbf{r} - \mathbf{U}^\top \mathbf{r}\right\|^2\right] \leq \epsilon \|\mathbf{r}\|^2.$$

Next, we turn to the subspace embedding property. The following result is remarkable because standard matrix tail bounds used to prove this property for leverage score sampling are not applicable to volume sampling. In fact, obtaining matrix Chernoff bounds for negatively associated joint distributions like volume sampling is an active area of research, as discussed in [21]. We address this challenge by defining a coupling procedure for volume sampling and uniform sampling without replacement, which leads to a curious reduction argument described in Appendix D.

**Theorem 9** *Let* $\mathbf{U} \in \mathbb{R}^{n \times d}$ *be a matrix s.t.* $\mathbf{U}^\top \mathbf{U} = \mathbf{I}$*. There is an absolute constant* $C$*, s.t. if sequence* $\pi \in [n]^k$ *is selected using leveraged volume sampling of size* $k \geq C \, d \ln(\frac{d}{\delta})$*, then*

$$\Pr\left(\lambda_{\min}\left(\frac{1}{k}\mathbf{U}^\top \mathbf{Q}_\pi \mathbf{U}\right) \leq \frac{1}{8}\right) \leq \delta.$$

Theorems 8 and 9 imply that the unbiased estimator $\mathbf{w}_\pi^*$ produced from leveraged volume sampling achieves multiplicative tail bounds with sample size $k = O(d \log d + d/\epsilon)$.

**Corollary 10** *Let* $\mathbf{X} \in \mathbb{R}^{n \times d}$ *be a full rank matrix. There is an absolute constant* $C$*, s.t. if sequence* $\pi \in [n]^k$ *is selected using leveraged volume sampling of size* $k \geq C \left(d \ln(\frac{d}{\delta}) + \frac{d}{\epsilon \delta}\right)$*, then for estimator*

$$\mathbf{w}_\pi^* = \underset{\mathbf{w}}{\operatorname{argmin}} \|\mathbf{Q}_\pi^{1/2}(\mathbf{X}\mathbf{w} - \mathbf{y})\|^2,$$

*we have* $L(\mathbf{w}_\pi^*) \leq (1 + \epsilon) \, L(\mathbf{w}^*)$ *with probability at least* $1 - \delta$*.*

**Proof** Let $\mathbf{U} = \mathbf{X}(\mathbf{X}^\top \mathbf{X})^{-1/2}$. Combining Theorem 8 with Markov's inequality, we have that for large enough $C$, $\|\mathbf{U}^\top \mathbf{Q}_\pi \mathbf{r}\|^2 \leq \epsilon \frac{k^2}{8^2} \|\mathbf{r}\|^2$ w.h.p., where $\mathbf{r} = \mathbf{y} - \mathbf{U}\mathbf{v}^*$. Finally following (6) and (7) above, we have that w.h.p.

$$L(\mathbf{w}_\pi^*) \leq L(\mathbf{w}^*) + \|(\mathbf{U}^\top \mathbf{Q}_\pi \mathbf{U})^{-1}\|^2 \, \|\mathbf{U}^\top \mathbf{Q}_\pi \mathbf{r}\|^2 \leq L(\mathbf{w}^*) + \frac{8^2}{k^2} \, \epsilon \frac{k^2}{8^2} \, \|\mathbf{r}\|^2 = (1 + \epsilon) \, L(\mathbf{w}^*). \; \blacksquare$$

## 5 Conclusion

We developed a new variant of volume sampling which produces the first known unbiased subsampled least-squares estimator with strong multiplicative loss bounds. In the process, we proved a novel extension of the Cauchy-Binet formula, as well as other fundamental combinatorial equalities. Moreover, we proposed an efficient algorithm called determinantal rejection sampling, which is to our knowledge the first joint determinantal sampling procedure that (after an initial $O(nd^2)$ preprocessing step for computing leverage scores) produces its $k$ samples in time $\widetilde{O}(d^2 + k)d^2)$, independent of the data size $n$. When $n$ is very large, the preprocessing time can be reduced to $\widetilde{O}(nd + d^5)$ by rescaling with sufficiently accurate approximations of the leverage scores. Surprisingly the estimator stays unbiased and the loss bound still holds with only slightly revised constants. For the sake of clarity we presented the algorithm based on rescaling with exact leverage scores in the main body of the paper. However we outline the changes needed when using approximate leverage scores in Appendix F.

In this paper we focused on tail bounds. However we conjecture that there are also volume sampling based unbiased estimators achieving expected loss bounds $\mathbb{E}[L(\mathbf{w}_\pi^*)] \leq (1+\epsilon)L(\mathbf{w}^*)$ with size $O(\frac{d}{\epsilon})$.

**Acknowledgements**

Michał Dereziński and Manfred K. Warmuth were supported by NSF grant IIS-1619271. Daniel Hsu was supported by NSF grant CCF-1740833.

## Footnotes

[1]The total loss being $1 + \epsilon$ times the optimum is the same as the regret being $\epsilon$ times the optimum.

[2]Otherwise just reduce $\mathbf{X}$ to a subset of independent columns. Also assume $\mathbf{X}$ has no rows of all zeros (every weight vector has the same loss on such rows, so they can be removed).

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
