[Supplementary Material]

# A Proof of (2) from Theorem 1

First, let us calculate $L(\mathbf{w}^*)$. Observe that

$$(\mathbf{X}^\top\mathbf{X})^{-1} = \overbrace{\left(1 + \frac{n-d}{d}\gamma^2\right)^{-1}}^{c}\mathbf{I},$$
$$\text{and}\quad \mathbf{w}^* = c\,\mathbf{X}^\top\mathbf{y} = c\,\mathbf{1}_d.$$

The loss $L(\mathbf{w})$ of any $\mathbf{w} \in \mathbb{R}^d$ can be decomposed as $L(\mathbf{w}) = \sum_{i=1}^d L_i(\mathbf{w})$, where $L_i(\mathbf{w})$ is the total loss incurred on all input vectors $\mathbf{e}_i$ or $\gamma\mathbf{e}_i$. For $\mathbf{w}^*$, the $i$-th component is

$$L_i(\mathbf{w}^*) = (1-c)^2 + \overbrace{\frac{n-d}{d}\gamma^2}^{\frac{1}{c}-1}c^2 = 1 - c.$$

Note that $i$-th leverage score of $\mathbf{X}$ is equal $l_i = \mathbf{x}_i^\top(\mathbf{X}^\top\mathbf{X})^{-1}\mathbf{x}_i = c$, so we obtain that

$$L(\mathbf{w}^*) = d\,(1-c) = \sum_{i=1}^d (1 - l_i). \tag{8}$$

Next, we compute $L(\mathbf{w}_S^*)$. Suppose that $S \subseteq \{1..n\}$ is produced by size $k$ standard volume sampling. Note that if for some $1 \le i \le d$ we have $i \notin S$, then $(\mathbf{w}_S^*)_i = 0$ and therefore $L_i(\mathbf{w}_S^*) = 1$. Moreover, denoting $b_i \overset{\text{def}}{=} \mathbf{1}_{[i \in S]}$,

$$(\mathbf{X}_S^\top\mathbf{X}_S)^{-1} \succeq (\mathbf{X}^\top\mathbf{X})^{-1} = c\,\mathbf{I}, \quad\text{and}\quad \mathbf{X}_S^\top\mathbf{y}_S = (b_1, \ldots, b_d)^\top,$$

so if $i \in S$, then $(\mathbf{w}_S^*)_i \ge c$ and

$$L_i(\mathbf{w}_S^*) \ge \frac{n-d}{d}\gamma^2\,c^2 = \left(\frac{1}{c} - 1\right)c^2 = c\,L_i(\mathbf{w}^*).$$

Putting the cases of $i \in S$ and $i \notin S$ together, we get

$$L_i(\mathbf{w}_S^*) \ge c\,L_i(\mathbf{w}^*) + (1 - c\,L_i(\mathbf{w}^*))\,(1 - b_i)$$
$$\ge c\,L_i(\mathbf{w}^*) + c^2(1 - b_i).$$

Applying the marginal probability formula for volume sampling (see (4)), we note that

$$\mathbb{E}[1 - b_i] = 1 - \Pr(i \in S) = \frac{n-k}{n-d}\,(1-c) = \frac{n-k}{n-d}\,L_i(\mathbf{w}^*).$$

Taking expectation over $L_i(\mathbf{w}_S^*)$ and summing the components over $i \in [d]$, we get

$$\mathbb{E}[L(\mathbf{w}_S^*)] \ge L(\mathbf{w}^*)\left(c + c^2\frac{n-k}{n-d}\right).$$

Note that as $\gamma \to 0$, we have $c \to 1$, thus showing (2).

# B Properties of rescaled volume sampling

We give proofs of the properties of rescaled volume sampling which hold for any rescaling distribution $q$. In this section, we will use $Z = d!\binom{k}{d}\det(\mathbf{X}^\top\mathbf{X})$ as the normalization constant for rescaled volume sampling.

## B.1 Proof of Proposition 2

First, we apply the Cauchy-Binet formula to the determinant term specified by a fixed sequence $\pi \in [n]^k$:

$$\det(\mathbf{X}^\top\mathbf{Q}_\pi\mathbf{X}) = \sum_{S \in \binom{[k]}{d}} \det(\mathbf{X}^\top\mathbf{Q}_{\pi_S}\mathbf{X}) = \sum_{S \in \binom{[k]}{d}} \det(\mathbf{X}_{\pi_S})^2 \prod_{i \in S} \frac{1}{q_{\pi_i}}.$$

Next, we compute the sum, using the above identity:

$$\sum_{\pi\in[n]^k}\det(\mathbf{X}^\top\mathbf{Q}_\pi\mathbf{X})\prod_{i=1}^k q_{\pi_i} = \sum_{\pi\in[n]^k}\sum_{S\in\binom{[k]}{d}}\det(\mathbf{X}_{\pi_S})^2\prod_{i\in[k]\setminus S} q_{\pi_i}$$

$$= \binom{k}{d}\sum_{\bar\pi\in[n]^d}\det(\mathbf{X}_{\bar\pi})^2\sum_{\tilde\pi\in[n]^{k-d}}\prod_{i=1}^{k-d} q_{\tilde\pi_i}$$

$$= \binom{k}{d}\sum_{\bar\pi\in[n]^d}\det(\mathbf{X}_{\bar\pi})^2\left(\sum_{i=1}^n q_i\right)^{k-d}$$

$$= \binom{k}{d}d!\sum_{S\in\binom{[n]}{d}}\det(\mathbf{X}_S)^2 = k(k-1)\cdots(k-d+1)\det(\mathbf{X}^\top\mathbf{X}),$$

where the steps closely follow the corresponding derivation for Theorem 3, given in Section 3.1.

## B.2   Proof of Theorem 4

We will prove that for any vector $\mathbf{v}\in\mathbb{R}^d$,

$$\mathbb{E}\big[\mathbf{v}^\top(\mathbf{X}^\top\mathbf{Q}_\pi\mathbf{X})^{-1}\mathbf{v}\big] \leq \frac{\mathbf{v}^\top(\mathbf{X}^\top\mathbf{X})^{-1}\mathbf{v}}{k-d+1},$$

which immediately implies the corresponding matrix inequality. First, we use Sylvester's formula, which holds whenever a matrix $\mathbf{A}\in\mathbb{R}^{d\times d}$ is full rank:

$$\det(\mathbf{A}+\mathbf{v}\mathbf{v}^\top) = \det(\mathbf{A})\left(1+\mathbf{v}^\top\mathbf{A}^{-1}\mathbf{v}\right).$$

Note that whenever the matrix is not full rank, its determinant is 0 (in which case we avoid computing the matrix inverse), so we have for any $\pi\in[n]^k$:

$$\det(\mathbf{X}^\top\mathbf{Q}_\pi\mathbf{X})\,\mathbf{v}^\top(\mathbf{X}^\top\mathbf{Q}_\pi\mathbf{X})^{-1}\mathbf{v} \leq \det(\mathbf{X}^\top\mathbf{Q}_\pi\mathbf{X}+\mathbf{v}\mathbf{v}^\top)-\det(\mathbf{X}^\top\mathbf{Q}_\pi\mathbf{X})$$

$$\overset{(*)}{=} \sum_{S\in\binom{[k]}{d-1}}\det(\mathbf{X}_{\pi_S}^\top\mathbf{X}_{\pi_S}+\mathbf{v}\mathbf{v}^\top)\prod_{i\in S}\frac{1}{q_{\pi_i}},$$

where $(*)$ follows from applying the Cauchy-Binet formula to both of the determinants, and cancelling out common terms. Next, we proceed in a standard fashion, summing over all $\pi\in[n]^k$:

$$Z\,\mathbb{E}\big[\mathbf{v}^\top(\mathbf{X}^\top\mathbf{Q}_\pi\mathbf{X})^{-1}\mathbf{v}\big] = \sum_{\pi\in[n]^k}\mathbf{v}^\top(\mathbf{X}^\top\mathbf{Q}_\pi\mathbf{X})^{-1}\mathbf{v}\det(\mathbf{X}^\top\mathbf{Q}_\pi\mathbf{X})\prod_{i=1}^k q_{\pi_i}$$

$$\leq \sum_{\pi\in[n]^k}\sum_{S\in\binom{[k]}{d-1}}\det(\mathbf{X}_{\pi_S}^\top\mathbf{X}_{\pi_S}+\mathbf{v}\mathbf{v}^\top)\prod_{i\in[k]\setminus S} q_{\pi_i}$$

$$= \binom{k}{d-1}\sum_{\bar\pi\in[n]^{d-1}}\det(\mathbf{X}_{\bar\pi}^\top\mathbf{X}_{\bar\pi}+\mathbf{v}\mathbf{v}^\top)\sum_{\tilde\pi\in[n]^{k-d+1}}\prod_{i=1}^{k-d+1} q_{\pi_i}$$

$$= \binom{k}{d-1}(d-1)!\sum_{S\in\binom{[n]}{d-1}}\det(\mathbf{X}_S^\top\mathbf{X}_S+\mathbf{v}\mathbf{v}^\top)$$

$$= \frac{d!\binom{k}{d}}{k-d+1}\big(\det(\mathbf{X}^\top\mathbf{X}+\mathbf{v}\mathbf{v}^\top)-\det(\mathbf{X}^\top\mathbf{X})\big) = Z\,\frac{\mathbf{v}^\top(\mathbf{X}^\top\mathbf{X})^{-1}\mathbf{v}}{k-d+1}.$$

## B.3 Proof of Proposition 5

First, we compute the marginal probability of a fixed element of sequence $\pi$ containing a particular index $i \in [n]$ under $q$-rescaled volume sampling:

$$Z \Pr(\pi_k = i) = \sum_{\pi \in [n]^{k-1}} \det(\mathbf{X}^\top \mathbf{Q}_{[\pi,i]} \mathbf{X}) \, q_i \prod_{t=1}^{k-1} q_{\pi_t}$$

$$= \underbrace{q_i \sum_{\pi \in [n]^{k-1}} \sum_{S \in \binom{[k-1]}{d}} \det(\mathbf{X}_{\pi_S})^2 \prod_{t \in [k-1] \setminus S} q_{\pi_t}}_{T_1} + \underbrace{\sum_{\pi \in [n]^{k-1}} \sum_{S \in \binom{[k-1]}{d-1}} \det(\mathbf{X}_{\pi_S}^\top \mathbf{X}_{\pi_S} + \mathbf{x}_i \mathbf{x}_i^\top) \prod_{t \in [k-1] \setminus S} q_{\pi_t}}_{T_2},$$

where the first term can be computed by following the derivation in Appendix B.1, obtaining $T_1 = q_i \frac{k-d}{k} Z$, and the second term is derived as in Appendix B.2, obtaining $T_2 = \frac{l_i}{k} Z$. Putting this together, we get

$$\Pr(\pi_k = i) = \frac{1}{k}\big((k-d)\, q_i + l_i\big).$$

Note that by symmetry this applies to any element of the sequence. We can now easily compute the desired expectation:

$$\mathbb{E}\big[(\mathbf{Q}_\pi)_{ii}\big] = \frac{1}{q_i} \sum_{t=1}^{k} \Pr(\pi_t = i) = (k-d) + \frac{l_i}{q_i}.$$

## B.4 Proof of Lemma 7

First step of the reverse iterative sampling procedure described in Section 2 involves removing one row from the given matrix with probability proportional to the square volume of that submatrix:

$$\forall_{i \in S} \qquad \Pr(i \mid \pi_S) = \frac{\det(\mathbf{X}^\top \mathbf{Q}_{\pi_{S \setminus i}} \mathbf{X})}{(|S| - d) \det(\mathbf{X}^\top \mathbf{Q}_{\pi_S} \mathbf{X})}.$$

Suppose that $k = s - 1$ and let $\tilde{\pi} = \pi_S \in [n]^{s-1}$ denote the sequence obtained after performing one step of the row-removal procedure. Then,

$$\Pr(\tilde{\pi}) = \sum_{\substack{\pi \in [n]^k: \\ \tilde{\pi} \text{ is a subsequence of } \pi}} \Pr(\tilde{\pi} \mid \pi) \Pr(\pi) \overset{(*)}{=} \sum_{i=1}^{n} s \overbrace{\Pr(i \mid [\tilde{\pi}, i])}^{\text{removing one row}} \overbrace{\Pr([\tilde{\pi}, i])}^{\text{rescaled sampling}}$$

$$= \sum_{i=1}^{n} s \frac{\det(\mathbf{X}^\top \mathbf{Q}_{\tilde{\pi}} \mathbf{X})}{(s-d) \det(\mathbf{X}^\top \mathbf{Q}_{[\tilde{\pi},i]} \mathbf{X})} \frac{\det(\mathbf{X}^\top \mathbf{Q}_{[\tilde{\pi},i]} \mathbf{X}) \left(\prod_{j=1}^{s-1} q_{\tilde{\pi}_j}\right) q_i}{\frac{s!}{(s-d)!} \det(\mathbf{X}^\top \mathbf{X})}$$

$$= \frac{\det(\mathbf{X}^\top \mathbf{Q}_{\tilde{\pi}} \mathbf{X})\left(\prod_{j=1}^{s-1} q_{\tilde{\pi}_j}\right)}{\frac{s-d}{s} \frac{s!}{(s-d)!} \det(\mathbf{X}^\top \mathbf{X})} \sum_{i=1}^{n} q_i = \frac{\det(\mathbf{X}^\top \mathbf{Q}_{\tilde{\pi}} \mathbf{X})\left(\prod_{j=1}^{s-1} q_{\tilde{\pi}_j}\right)}{\frac{(s-1)!}{(s-1-d)!} \det(\mathbf{X}^\top \mathbf{X})},$$

where $(*)$ follows because the ordering of sequence $\pi$ does not affect the probabilities, and the factor $s$ next to the sum counts the number of ways to place index $i$ into the sequence $\tilde{\pi}$ to obtain $\pi$. Thus, by induction, for any $k < s$ the algorithm correctly samples from $q$-rescaled volume sampling.

## C  Proof of Theorem 8

We rewrite the expected square norm as:

$$\mathbb{E}\left[\left\|\frac{1}{k}\mathbf{U}^\top \mathbf{Q}_\pi \mathbf{r} - \mathbf{U}^\top \mathbf{r}\right\|^2\right] = \mathbb{E}\left[\left\|\mathbf{U}^\top\left(\frac{1}{k}\mathbf{Q}_\pi - \mathbf{I}\right)\mathbf{r}\right\|^2\right] = \mathbb{E}\left[\mathbf{r}^\top\left(\frac{1}{k}\mathbf{Q}_\pi - \mathbf{I}\right)\mathbf{U}\mathbf{U}^\top\left(\frac{1}{k}\mathbf{Q}_\pi - \mathbf{I}\right)\mathbf{r}\right]$$

$$= \mathbf{r}^\top \mathbb{E}\left[\left(\frac{1}{k}\mathbf{Q}_\pi - \mathbf{I}\right)\mathbf{U}\mathbf{U}^\top\left(\frac{1}{k}\mathbf{Q}_\pi - \mathbf{I}\right)\right]\mathbf{r}$$

$$\leq \lambda_{\max}\Big(\underbrace{\big(\mathbb{E}[(z_i - 1)(z_j - 1)]\,\mathbf{u}_i^\top \mathbf{u}_j\big)_{ij}}_{\mathbf{M}}\Big)\|\mathbf{r}\|^2, \quad \text{where } z_i = \frac{1}{k}(\mathbf{Q}_\pi)_{ii}.$$

It remains to bound $\lambda_{\max}(\mathbf{M})$. By Proposition 5, for leveraged volume sampling $\mathbb{E}[(\mathbf{Q}_\pi)_{ii}] = k$, so

$$\mathbb{E}[(z_i-1)(z_j-1)] = \frac{1}{k^2}\Big(\mathbb{E}\big[(\mathbf{Q}_\pi)_{ii}(\mathbf{Q}_\pi)_{jj}\big] - \mathbb{E}[(\mathbf{Q}_\pi)_{ii}]\,\mathbb{E}[(\mathbf{Q}_\pi)_{jj}]\Big) = \frac{1}{k^2}\,\mathrm{cov}\big[(\mathbf{Q}_\pi)_{ii},\,(\mathbf{Q}_\pi)_{jj}\big].$$

For rescaled volume sampling this is given in the following lemma, proven in Appendix C.1.

**Lemma 11** *For any* $\mathbf{X}$ *and* $q$*, if sequence* $\pi \in [n]^k$ *is sampled from* $q$*-rescaled volume sampling then*

$$\mathrm{cov}\big[(\mathbf{Q}_\pi)_{ii},\,(\mathbf{Q}_\pi)_{jj}\big] = \mathbf{1}_{i=j}\frac{1}{q_i}\mathbb{E}[(\mathbf{Q}_\pi)_{ii}] - (k-d) - \frac{(\mathbf{x}_i^\top(\mathbf{X}^\top\mathbf{X})^{-1}\mathbf{x}_j)^2}{q_iq_j}.$$

Since $\|\mathbf{u}_i\|^2 = l_i = dq_i$ and $\mathbf{u}_i^\top(\mathbf{U}^\top\mathbf{U})^{-1}\mathbf{u}_j = \mathbf{u}_i^\top\mathbf{u}_j$, we can express matrix $\mathbf{M}$ as follows:

$$\mathbf{M} = \mathrm{diag}\left(\frac{d\,\mathbb{E}[(\mathbf{Q}_\pi)_{ii}]}{\|\mathbf{u}_i\|^2k^2}\|\mathbf{u}_i\|^2\right)_{i=1}^n - \frac{k-d}{k^2}\mathbf{U}\mathbf{U}^\top - \frac{d^2}{k^2}\left(\frac{(\mathbf{u}_i^\top\mathbf{u}_j)^3}{\|\mathbf{u}_i\|^2\|\mathbf{u}_j\|^2}\right)_{ij}.$$

The first term simplifies to $\frac{d}{k}\mathbf{I}$, and the second term is negative semi-definite, so

$$\lambda_{\max}(\mathbf{M}) \le \frac{d}{k} + \frac{d^2}{k^2}\left\|\left(\frac{(\mathbf{u}_i^\top\mathbf{u}_j)^3}{\|\mathbf{u}_i\|^2\|\mathbf{u}_j\|^2}\right)_{ij}\right\|.$$

Finally, we decompose the last term into a Hadamard product of matrices, and apply a classical inequality by [3] (symbol "∘" denotes Hadamard matrix product—i.e., elementwise multiplication):

$$\begin{aligned}
\left\|\left(\frac{(\mathbf{u}_i^\top\mathbf{u}_j)^3}{\|\mathbf{u}_i\|^2\|\mathbf{u}_j\|^2}\right)_{ij}\right\| &= \left\|\left(\frac{\mathbf{u}_i^\top\mathbf{u}_j}{\|\mathbf{u}_i\|\,\|\mathbf{u}_j\|}\right)_{ij}\circ\left(\frac{(\mathbf{u}_i^\top\mathbf{u}_j)^2}{\|\mathbf{u}_i\|\|\mathbf{u}_j\|}\right)_{ij}\right\| \\
&\le \left\|\left(\frac{(\mathbf{u}_i^\top\mathbf{u}_j)^2}{\|\mathbf{u}_i\|\|\mathbf{u}_j\|}\right)_{ij}\right\| = \left\|\left(\frac{\mathbf{u}_i^\top\mathbf{u}_j}{\|\mathbf{u}_i\|\,\|\mathbf{u}_j\|}\right)_{ij}\circ\mathbf{U}\mathbf{U}^\top\right\| \\
&\le \|\mathbf{U}\mathbf{U}^\top\| = 1.
\end{aligned}$$

Thus, we conclude that $\mathbb{E}[\|\frac{1}{k}\mathbf{U}^\top\mathbf{Q}_\pi\mathbf{r} - \mathbf{U}^\top\mathbf{r}\|^2] \le (\frac{d}{k} + \frac{d^2}{k^2})\|\mathbf{r}\|^2$, completing the proof.

### C.1   Proof of Lemma 11

We compute marginal probability of two elements in the sequence $\pi$ having particular values $i, j \in [n]$:

$$Z\,\mathrm{Pr}\big((\pi_{k-1}=i)\wedge(\pi_k=j)\big) = \sum_{\pi\in[n]^{k-2}}\sum_{S\in\binom{[k]}{d}}\det(\mathbf{X}_{[\pi,i,j]_S}^\top\mathbf{X}_{[\pi,i,j]_S})\prod_{t\in[k]\backslash S}q_{[\pi,i,j]_t}.$$

We partition the set $\binom{[k]}{d}$ of all subsets of size $d$ into four groups, and summing separately over each of the groups, we have

$$Z\,\mathrm{Pr}\big((\pi_{k-1}=i)\wedge(\pi_k=j)\big) = T_{00} + T_{01} + T_{10} + T_{11}, \qquad \text{where:}$$

1. Let $G_{00} = \{S\in\binom{[k]}{d} : k-1\notin S,\ k\notin S\}$, and following derivation in Appendix B.1,

$$T_{00} = q_i\,q_j\sum_{\pi\in[n]^{k-2}}\sum_{S\in G_{00}}\det(\mathbf{X}_{\pi_S})^2\prod_{t\in[k-2]\backslash S}q_{\pi_t} = q_i\,q_j\frac{(k-d-1)(k-d)}{(k-1)\,k}\,Z.$$

2. Let $G_{10} = \{S\in\binom{[k]}{d} : k-1\in S,\ k\notin S\}$, and following derivation in Appendix B.2,

$$T_{10} = q_j\sum_{\pi\in[n]^{k-1}}\sum_{S\in G_{10}}\det(\mathbf{X}_{[\pi,i]_S})^2\prod_{t\in[k-1]\backslash S}q_{[\pi,i]_t} = l_i\,q_j\frac{(k-d)}{(k-1)\,k}\,Z.$$

3. $G_{01} = \{S\in\binom{[k]}{d} : k-1\notin S,\ k\in S\}$, and by symmetry, $T_{01} = l_j\,q_i\frac{(k-d)}{(k-1)\,k}\,Z$.

4. Let $G_{11} = \{S \in \binom{[k]}{d} : \ k{-}1 \in S, \ k \in S\}$, and the last term is

$$
\begin{aligned}
T_{11} &= \sum_{\pi \in [n]^{k-1}} \sum_{S \in G_{11}} \det(\mathbf{X}_{[\pi,i,j]_S})^2 \prod_{t \in [k] \setminus S} q_{[\pi,i,j]_t} \\
&= \binom{k{-}2}{d{-}2} \sum_{\pi \in [n]^{d-2}} \det(\mathbf{X}_{[\pi,i,j]})^2 \\
&= \binom{k{-}2}{d{-}2} (d{-}2)! \left( \det(\mathbf{X}^\top \mathbf{X}) - \det(\mathbf{X}_{-i}^\top \mathbf{X}_{-i}) - \det(\mathbf{X}_{-j}^\top \mathbf{X}_{-j}) + \det(\mathbf{X}_{-i,j}^\top \mathbf{X}_{-i,j}) \right) \\
&\overset{(*)}{=} \frac{d! \binom{k}{d}}{k(k{-}1)} \det(\mathbf{X}^\top \mathbf{X}) \Big( 1 - \underbrace{(1{-}l_i)}_{\frac{\det(\mathbf{X}_{-i}^\top \mathbf{X}_{-i})}{\det(\mathbf{X}^\top \mathbf{X})}} - \underbrace{(1{-}l_j)}_{\frac{\det(\mathbf{X}_{-j}^\top \mathbf{X}_{-j})}{\det(\mathbf{X}^\top \mathbf{X})}} + \underbrace{(1{-}l_i)(1{-}l_j) - l_{ij}^2}_{\frac{\det(\mathbf{X}_{-i,j}^\top \mathbf{X}_{-i,j})}{\det(\mathbf{X}^\top \mathbf{X})}} \Big) \\
&= \frac{Z}{k(k{-}1)} \big( \ell_i \ell_j - \ell_{ij}^2 \big),
\end{aligned}
$$

where $l_{ij} = \mathbf{x}_i^\top (\mathbf{X}^\top \mathbf{X})^{-1} \mathbf{x}_j$, and $(*)$ follows from repeated application of Sylvester's determinant formula (as in Appendix B.2). Putting it all together, we can now compute the expectation for $i \neq j$:

$$
\begin{aligned}
\mathbb{E}\big[ (\mathbf{Q}_\pi)_{ii} (\mathbf{Q}_\pi)_{jj} \big] &= \frac{1}{q_i\, q_j} \sum_{t_1=1}^{k} \sum_{t_2=1}^{k} \Pr\big( (\pi_{k-1}{=}i) \wedge (\pi_k{=}j) \big) \\
&= \frac{k(k{-}1)}{q_i\, q_j} \frac{\overbrace{\frac{1}{Z}(T_{00}+T_{10}+T_{01}+T_{11})}}{\Pr\big( (\pi_{k-1}{=}i) \wedge (\pi_k{=}j) \big)} \\
&= (k{-}d{-}1)(k{-}d) + (k{-}d)\frac{l_i}{q_i} + (k{-}d)\frac{l_j}{q_j} + \frac{l_i l_j}{q_i\, q_j} - \frac{l_{ij}^2}{q_i\, q_j} \\
&= \left( (k{-}d)q_i + \frac{l_i}{q_i} \right) \left( (k{-}d)q_j + \frac{l_j}{q_j} \right) - (k{-}d) - \frac{l_{ij}^2}{q_i\, q_j} \\
&= \mathbb{E}\big[ (\mathbf{Q}_\pi)_{ii} \big]\, \mathbb{E}\big[ (\mathbf{Q}_\pi)_{jj} \big] - (k{-}d) - \frac{l_{ij}^2}{q_i q_j}.
\end{aligned}
$$

Finally, if $i = j$, then

$$
\begin{aligned}
\mathbb{E}\big[ (\mathbf{Q}_\pi)_{ii} (\mathbf{Q}_\pi)_{ii} \big] &= \frac{1}{q_i^2} \sum_{t_1=1}^{k} \sum_{t_2=1}^{k} \Pr(\pi_{t_1}{=}i \wedge \pi_{t_2}{=}i) \\
&= \frac{k(k{-}1)}{q_i^2} \Pr(\pi_{k-1}{=}i \wedge \pi_k{=}i) + \frac{k}{q_i^2} \Pr(\pi_k{=}i) \\
&= \big( \mathbb{E}\big[ (\mathbf{Q}_\pi)_{ii} \big] \big)^2 - (k{-}d) - \frac{l_i^2}{q_i^2} + \frac{1}{q_i} \mathbb{E}\big[ (\mathbf{Q}_\pi)_{ii} \big].
\end{aligned}
$$

# D  Proof of Theorem 9

We break the sampling procedure down into two stages. First, we do leveraged volume sampling of a sequence $\pi \in [n]^m$ of size $m \geq C_0 d^2/\delta$, then we do standard volume sampling size $k$ from matrix $(\mathbf{Q}_{[1..n]}^{1/2} \mathbf{U})_\pi$. Since rescaled volume sampling is closed under this subsampling (Lemma 7), this procedure is equivalent to size $k$ leveraged volume sampling from $\mathbf{U}$. To show that the first stage satisfies the subspace embedding condition, we simply use the bound from Theorem 8 (see details in Appendix D.1):

**Lemma 12** *There is an absolute constant $C_0$, s.t. if sequence $\pi \in [n]^m$ is generated via leveraged volume sampling of size $m$ at least $C_0\, d^2/\delta$ from $\mathbf{U}$, then*

$$
\Pr\left( \lambda_{\min}\Big( \frac{1}{m} \mathbf{U}^\top \mathbf{Q}_\pi \mathbf{U} \Big) \leq \frac{1}{2} \right) \leq \delta.
$$

The size of $m$ is much larger than what we claim is sufficient. However, we use it to achieve a tighter bound in the second stage. To obtain substantially smaller sample sizes for subspace embedding than what Theorem 8 can deliver, it is standard to use tail bounds for the sums of independent matrices. However, applying these results to joint sampling is a challenging task. Interestingly, [26] showed that volume sampling is a strongly Raleigh measure, implying that the sampled vectors are negatively correlated. This guarantee is sufficient to show tail bounds for real-valued random variables [see, e.g., 30], however it has proven challenging in the matrix case, as discussed by [21]. One notable exception is uniform sampling without replacement, which is a negatively correlated joint distribution. A reduction argument originally proposed by [22], but presented in this context by [20], shows that uniform sampling without replacement offers the same tail bounds as i.i.d. uniform sampling.

**Lemma 13** *Assume that* $\lambda_{\min}\left(\frac{1}{m}\mathbf{U}^\top\mathbf{Q}_\pi\mathbf{U}\right) \geq \frac{1}{2}$. *Suppose that set* $T$ *is a set of fixed size sampled uniformly without replacement from* $[m]$. *There is a constant* $C_1$ *s.t. if* $|T| \geq C_1\,d\ln(d/\delta)$, *then*

$$\Pr\left(\lambda_{\min}\left(\frac{1}{|T|}\mathbf{U}^\top\mathbf{Q}_{\pi_T}\mathbf{U}\right) \leq \frac{1}{4}\right) \leq \delta.$$

The proof of Lemma 13 (given in appendix D.2) is a straight-forward application of the argument given by [20]. We now propose a different reduction argument showing that a subspace embedding guarantee for uniform sampling without replacement leads to a similar guarantee for volume sampling. We achieve this by exploiting a volume sampling algorithm proposed recently by [13], shown in Algorithm 3, which is a modification of the reverse iterative sampling procedure introduced in [11]. This procedure relies on iteratively removing elements from the set $S$ until we are left with $k$ elements. Specifically, at each step, we sample an index $i$ from a conditional distribution, $i \sim \Pr(i\,|\,S) = (1 - \frac{1}{q_{\pi_i}}\mathbf{u}_{\pi_i}^\top(\mathbf{U}^\top\mathbf{Q}_{\pi_S}\mathbf{U})^{-1}\mathbf{u}_{\pi_i})/(|S| - d)$. Crucially for us, each step proceeds via rejection sampling with the proposal distribution being uniform. We can easily modify the algorithm, so that the samples from the proposal distribution are used to construct a uniformly sampled set $T$, as shown in Algorithm 4. Note that sets $S$ returned by both algorithms are identically distributed, and furthermore, $T$ is a subset of $S$, because every index taken out of $S$ is also taken out of $T$.

<div style="display:flex">

**Algorithm 3: Volume sampling**
1: $S \leftarrow [m]$
2: **while** $|S| > k$
3:     **repeat**
4:       Sample $i$ unif. out of $S$
5:       $p \leftarrow 1 - \frac{1}{q_{\pi_i}}\mathbf{u}_{\pi_i}^\top(\mathbf{U}^\top\mathbf{Q}_{\pi_S}\mathbf{U})^{-1}\mathbf{u}_{\pi_i}$
6:       Sample $Accept \sim$ Bernoulli$(p)$
7:     **until** $Accept =$ true
8:     $S \leftarrow S\backslash\{i\}$
9: **end**
10: **return** $S$

**Algorithm 4: Coupled sampling**
1: $S, T \leftarrow [m]$
2: **while** $|S| > k$
3:     Sample $i$ unif. out of $[m]$
4:     $T \leftarrow T - \{i\}$
5:     **if** $i \in S$
6:       $p \leftarrow 1 - \frac{1}{q_{\pi_i}}\mathbf{u}_{\pi_i}^\top(\mathbf{U}^\top\mathbf{Q}_{\pi_S}\mathbf{U})^{-1}\mathbf{u}_{\pi_i}$
7:       Sample $Accept \sim$ Bernoulli$(p)$
8:       **if** $Accept =$ true,    $S \leftarrow S\backslash\{i\}$ **end**
9:     **end**
10: **end**
11: **return** $S, T$

</div>

By Lemma 13, if size of $T$ is at least $C_1\,d\log(d/\delta)$, then this set offers a subspace embedding guarantee. Next, we will show that in fact set $T$ is not much smaller than $S$, implying that the same guarantee holds for $S$. Specifically, we will show that $|S \setminus T| = O(d\log(d/\delta))$. Note that it suffices to bound the number of times that a uniform sample is rejected by sampling $A = 0$ in line 7 of Algorithm 4. Denote this number by $R$. Note that $R = \sum_{t=k+1}^m R_t$, where $m = |Q|$ and $R_t$ is the number of times that $A = 0$ was sampled while the size of set $S$ was $t$. Variables $R_t$ are independent, and each is distributed according to the geometric distribution (number of failures until success), with the success probability

$$r_t = \frac{1}{t}\sum_{i \in S}\left(1 - \frac{1}{q_{\pi_i}}\mathbf{u}_{\pi_i}^\top(\mathbf{U}^\top\mathbf{Q}_{\pi_S}\mathbf{U})^{-1}\mathbf{u}_{\pi_i}\right) = \frac{1}{t}\left(t - \mathrm{tr}\left((\mathbf{U}^\top\mathbf{Q}_{\pi_S}\mathbf{U})^{-1}\mathbf{U}^\top\mathbf{Q}_{\pi_S}\mathbf{U}\right)\right) = \frac{t-d}{t}.$$

Now, as long as $\frac{m-d}{k-d} \leq C_0\,d^2/\delta$, we can bound the expected value of $R$ as follows:

$$\mathbb{E}[R] = \sum_{t=k+1}^m \mathbb{E}[R_t] = \sum_{t=k+1}^m \left(\frac{t}{t-d} - 1\right) = d\sum_{t=k-d+1}^{m-d}\frac{1}{t} \leq d\,\ln\left(\frac{m-d}{k-d}\right) \leq C_2\,d\ln(d/\delta).$$

In this step, we made use of the first stage sampling, guaranteeing that the term under the logarithm is bounded. Next, we show that the upper tail of $R$ decays very rapidly given a sufficiently large gap between $m$ and $k$ (proof in Appendix D.3):

**Lemma 14** *Let $R_t \sim \mathrm{Geom}(\frac{t-d}{t})$ be a sequence of independent geometrically distributed random variables (number of failures until success). Then, for any $d < k < m$ and $a > 1$,*

$$\Pr\big(R \geq a\,\mathbb{E}[R]\big) \leq e^{\frac{a}{2}} \left(\frac{k-d}{m-d}\right)^{\frac{a}{2}-1} \quad for \quad R = \sum_{t=k+1}^{m} R_t.$$

Let $a = 4$ in Lemma 14. Setting $C = C_1 + 2a\,C_2$, for any $k \geq C\,d\ln(d/\delta)$, using $m = \max\{C_0\,\frac{d^2}{\delta},\ d + \mathrm{e}^2\,\frac{k}{\delta}\}$, we obtain that

$$R \leq a\,C_2\,d\ln(d/\delta) \leq k/2, \quad \text{w.p.} \quad \geq 1 - \mathrm{e}^2\,\frac{k-d}{m-d} \geq 1 - \delta,$$

showing that $|T| \geq k - R \geq C_1\,d\ln(d/\delta)$ and $k \leq 2|T|$.

Therefore, by Lemmas 12, 13 and 14, there is a $1 - 3\delta$ probability event in which

$$\lambda_{\min}\left(\frac{1}{|T|}\mathbf{U}^\top\mathbf{Q}_{\pi_T}\mathbf{U}\right) \geq \frac{1}{4} \quad \text{and} \quad k \leq 2|T|.$$

In this same event,

$$\lambda_{\min}\left(\frac{1}{k}\mathbf{U}^\top\mathbf{Q}_{\pi_S}\mathbf{U}\right) \geq \lambda_{\min}\left(\frac{1}{k}\mathbf{U}^\top\mathbf{Q}_{\pi_T}\mathbf{U}\right) \geq \lambda_{\min}\left(\frac{1}{2|T|}\mathbf{U}^\top\mathbf{Q}_{\pi_T}\mathbf{U}\right) \geq \frac{1}{2}\cdot\frac{1}{4} = \frac{1}{8},$$

which completes the proof of Theorem 9.

### D.1 Proof of Lemma 12

Replacing vector $\mathbf{r}$ in Theorem 8 with each column of matrix $\mathbf{U}$, we obtain that for $m \geq C\,\frac{d}{\epsilon}$,

$$\mathbb{E}\big[\|\mathbf{U}^\top\mathbf{Q}_\pi\mathbf{U} - \mathbf{U}^\top\mathbf{U}\|_F^2\big] \leq \epsilon\,\|\mathbf{U}\|_F^2 = \epsilon\,d.$$

We bound the 2-norm by the Frobenius norm and use Markov's inequality, showing that w.p. $\geq 1 - \delta$

$$\|\mathbf{U}^\top\mathbf{Q}_\pi\mathbf{U} - \mathbf{I}\| \leq \|\mathbf{U}^\top\mathbf{Q}_\pi\mathbf{U} - \mathbf{I}\|_F \leq \sqrt{\epsilon\,d/\delta}.$$

Setting $\epsilon = \frac{\delta}{4d}$, for $m \geq C_0\,d^2/\delta$, the above inequality implies that

$$\lambda_{\min}\left(\frac{1}{m}\mathbf{U}^\top\mathbf{Q}_\pi\mathbf{U}\right) \geq \frac{1}{2}.$$

### D.2 Proof of Lemma 13

Let $\pi$ denote the sequence of $m$ indices selected by volume sampling in the first stage. Suppose that $i_1, ..., i_k$ are independent uniformly sampled indices from $[m]$, and let $j_1, ..., j_k$ be indices sampled uniformly without replacement from $[m]$. We define matrices

$$\mathbf{Z} \overset{def}{=} \sum_{t=1}^{k} \overbrace{\frac{1}{kq_{i_t}}\mathbf{u}_{i_t}\mathbf{u}_{i_t}^\top}^{\mathbf{z}_t}, \quad \text{and} \quad \widehat{\mathbf{Z}} \overset{def}{=} \sum_{t=1}^{k} \overbrace{\frac{1}{kq_{j_t}}\mathbf{u}_{j_t}\mathbf{u}_{j_t}^\top}^{\widehat{\mathbf{z}}_t}.$$

Note that $\|\mathbf{Z}_t\| = \frac{d}{k\,l_i}\|\mathbf{u}_{i_t}\|^2 = \frac{d}{k}$ and, similarly, $\|\widehat{\mathbf{Z}}_t\| = \frac{d}{k}$. Moreover,

$$\mathbb{E}[\mathbf{Z}] = \sum_{t=1}^{k}\left[\frac{1}{m}\sum_{i=1}^{m}\frac{1}{kq_i}\mathbf{u}_i\mathbf{u}_i^\top\right] = k\,\frac{1}{k}\frac{1}{m}\mathbf{U}^\top\mathbf{Q}_\pi\mathbf{U} = \frac{1}{m}\mathbf{U}^\top\mathbf{Q}_\pi\mathbf{U}.$$

Combining Chernoff's inequality with the reduction argument described in [20], for any $\lambda$, and $\theta > 0$,

$$\Pr\big(\lambda_{\max}(-\widehat{\mathbf{Z}}) \geq \lambda\big) \leq \mathrm{e}^{-\theta\lambda}\,\mathbb{E}\Big[\mathrm{tr}\big(\exp(\theta(-\widehat{\mathbf{Z}}))\big)\Big] \leq \mathrm{e}^{-\theta\lambda}\,\mathbb{E}\Big[\mathrm{tr}\big(\exp(\theta(-\mathbf{Z}))\big)\Big].$$

Using matrix Chernoff bound of [32] applied to $-\mathbf{Z}_1, ..., -\mathbf{Z}_k$ with appropriate $\theta$, we have

$$\mathrm{e}^{-\theta\lambda}\,\mathbb{E}\Big[\mathrm{tr}\big(\exp(\theta(-\mathbf{Z}))\big)\Big] \leq d\,\exp\left(-\frac{k}{16d}\right), \quad \text{for} \quad \lambda = \frac{1}{2}\lambda_{\max}\left(-\frac{1}{m}\mathbf{U}^\top\mathbf{Q}_\pi\mathbf{U}\right) \leq -\frac{1}{4}.$$

Thus, there is a constant $C_1$ such that for $k \geq C_1\,d\ln(d/\delta)$, w.p. at least $1 - \delta$ we have $\lambda_{\min}(\widehat{\mathbf{Z}}) \geq \frac{1}{4}$.

## D.3 Proof of Lemma 14

We compute the moment generating function of the variable $R_t \sim \text{Geom}(r_t)$, where $r_t = \frac{t-d}{t}$:

$$\mathbb{E}\big[e^{\theta R_t}\big] = \frac{r_t}{1 - (1 - r_t)e^\theta} = \frac{\frac{t-d}{t}}{1 - \frac{d}{t}\,e^\theta} = \frac{t-d}{t - d\,e^\theta}.$$

Setting $\theta = \frac{1}{2d}$, we observe that $de^\theta \leq d + 1$, and so $\mathbb{E}[e^{\theta R_t}] \leq \frac{t-d}{t-d-1}$. Letting $\mu = \mathbb{E}[R]$, for any $a > 1$ using Markov's inequality we have

$$\Pr(R \geq a\mu) \leq e^{-a\theta\mu}\,\mathbb{E}\big[e^{\theta R}\big] \leq e^{-a\theta\mu} \prod_{t=k+1}^{m} \frac{t-d}{t-d-1} = e^{-a\theta\mu}\,\frac{m-d}{k-d}.$$

Note that using the bounds on the harmonic series we can estimate the mean:

$$\mu = d \sum_{t=k-d+1}^{m-d} \frac{1}{t} \geq d\left(\ln(m-d) - \ln(k-d) - 1\right) = d\ln\left(\frac{m-d}{k-d}\right) - d,$$

$$\text{so}\quad e^{-a\theta\mu} \leq e^{a/2}\exp\left(-\frac{a}{2}\ln\left(\frac{m-d}{k-d}\right)\right) = e^{a/2}\left(\frac{m-d}{k-d}\right)^{-a/2}.$$

Putting the two inequalities together we obtain the desired tail bound.

## E  Experiments

We present experiments comparing leveraged volume sampling to standard volume sampling and to leverage score sampling, in terms of the total square loss suffered by the subsampled least-squares estimator. The three estimators can be summarized as follows:

$$\textit{volume sampling:}\quad \mathbf{w}_S^* = (\mathbf{X}_S)^+\mathbf{y}_S, \qquad \Pr(S) \sim \det(\mathbf{X}_S^\top\mathbf{X}_S), \quad S \in \binom{[n]}{k};$$

$$\textit{leverage score sampling:}\quad \mathbf{w}_\pi^* = (\mathbf{Q}_\pi^{1/2}\mathbf{X})^+\mathbf{Q}_\pi^{1/2}\mathbf{y}, \quad \Pr(\pi) = \prod_{i=1}^{k} \frac{l_{\pi_i}}{d}, \qquad \pi \in [n]^k;$$

$$\textit{leveraged volume sampling:}\quad \mathbf{w}_\pi^* = (\mathbf{Q}_\pi^{1/2}\mathbf{X})^+\mathbf{Q}_\pi^{1/2}\mathbf{y}, \quad \Pr(\pi) \sim \det(\mathbf{X}^\top\mathbf{Q}_\pi\mathbf{X}) \prod_{i=1}^{k} \frac{l_{\pi_i}}{d}.$$

Both the volume sampling-based estimators are unbiased, however the leverage score sampling estimator is not. Recall that $\mathbf{Q}_\pi = \sum_{i=1}^{|\pi|} q_{\pi_i}^{-1}\mathbf{e}_{\pi_i}\mathbf{e}_{\pi_i}^\top$ is the selection and rescaling matrix as defined for $q$-rescaled volume sampling with $q_i = \frac{l_i}{d}$. For each estimator we plotted its average total loss, i.e., $\frac{1}{n}\|\mathbf{X}\mathbf{w} - \mathbf{y}\|^2$, for a range of sample sizes $k$, contrasted with the loss of the best least-squares estimator $\mathbf{w}^*$ computed from all data.

| Dataset | Instances ($n$) | Features ($d$) |
|---------|-----------------|----------------|
| *bodyfat* | 252 | 14 |
| *housing* | 506 | 13 |
| *mg* | 1,385 | 21 |
| *abalone* | 4,177 | 36 |
| *cpusmall* | 8,192 | 12 |
| *cadata* | 20,640 | 8 |
| *MSD* | 463,715 | 90 |

Table 1: Libsvm regression datasets [9] (to increase dimensionality of *mg* and *abalone*, we expanded features to all degree 2 monomials, and removed redundant ones).

Plots shown in Figures 1 and 2 were averaged over 100 runs, with shaded area representing standard error of the mean. We used seven benchmark datasets from the libsvm repository [9] (six in this section and one in Section 1), whose dimensions are given in Table 1. The results confirm that leveraged volume sampling is as good or better than either of the baselines for any sample size $k$. We can see that in some of the examples standard volume sampling exhibits bad behavior for larger sample sizes, as suggested by the lower bound of Theorem 1 (especially noticeable on *bodyfat* and *cpusmall* datasets). On the other hand, leverage score sampling exhibits poor performance for small sample sizes due to the coupon collector problem, which is most noticeable for *abalone* dataset, where we can see a very sharp transition after which leverage score sampling becomes effective. Neither of the variants of volume sampling suffers from this issue.

Figure 2: Comparison of loss of the subsampled estimator when using *leveraged volume sampling* vs using *leverage score sampling* and standard *volume sampling* on six datasets.

## F   Faster algorithm via approximate leverage scores

In some settings, the primary computational cost of deploying leveraged volume sampling is the preprocessing cost of computing exact laverage scores for matrix $\mathbf{X} \in \mathbb{R}^{n \times d}$, which takes $O(nd^2)$. There is a large body of work dedicated to fast estimation of leverage scores (see, e.g., [16, 27]), and in this section we examine how these approaches can be utilized to make leveraged volume sampling more efficient. The key challenge here is to show that the determinantal rejection sampling step remains effective when distribution $q$ consists of approximate leverage scores. Our strategy, which is described in the algorithm *fast leveraged volume sampling*, will be to compute an approximate covariance matrix $\mathbf{A} = (1 \pm \epsilon)\mathbf{X}^\top \mathbf{X}$ and use it to compute the rescaling distribution $q_i \sim \mathbf{x}_i^\top \mathbf{A}^{-1} \mathbf{x}_i$. As we see in the lemma below, for sufficiently small $\epsilon$, this rescaling still retains the runtime guarantee of determinantal rejection sampling from Theorem 6.

---
**Fast leveraged volume sampling**

**Input:** $\mathbf{X} \in \mathbb{R}^{n \times d}$, $k \geq d$, $\epsilon \geq 0$
Compute $\mathbf{A} = (1 \pm \epsilon)\,\mathbf{X}^\top \mathbf{X}$
Compute $\tilde{l}_i = (1 \pm \frac{1}{2})\,l_i \quad \forall_{i \in [n]}$
$s \leftarrow \max\{k,\, 8d^2\}$
**repeat**
  $\pi \leftarrow$ empty sequence
  **while** $|\pi| < s$
    Sample $i \sim (\tilde{l}_1, \ldots, \tilde{l}_n)$
    $a \sim \text{Bernoulli}\left((1 - \epsilon)\frac{\mathbf{x}_i^\top \mathbf{A}^{-1}\mathbf{x}_i}{2\tilde{l}_i}\right)$
    **if** $a = $ true, **then** $\pi \leftarrow [\pi, i]$
  **end**
  $\mathbf{Q}_\pi \leftarrow \sum_{j=1}^{s} d\,(\mathbf{x}_{\pi_j}^\top \mathbf{A}^{-1}\mathbf{x}_{\pi_j})^{-1}\mathbf{e}_{\pi_j}\mathbf{e}_{\pi_j}^\top$
  Sample $Acc \sim \text{Bernoulli}\left(\frac{\det(\frac{1}{s}\mathbf{X}^\top \mathbf{Q}_\pi \mathbf{X})}{\det(\mathbf{A})}\right)$
**until** $Acc = $ true
$S \leftarrow \text{VolumeSample}\left((\mathbf{Q}_{[1..n]}^{1/2}\mathbf{X})_\pi, k\right)$
**return** $\pi_S$

---

**Lemma 15** *Let* $\mathbf{X} \in \mathbb{R}^{n \times d}$ *be a full rank matrix, and suppose that matrix* $\mathbf{A} \in \mathbb{R}^{d \times d}$ *satisfies*

$$(1 - \epsilon)\,\mathbf{X}^\top \mathbf{X} \preceq \mathbf{A} \preceq (1 + \epsilon)\,\mathbf{X}^\top \mathbf{X}, \quad \text{where} \quad \frac{\epsilon}{1 - \epsilon} \leq \frac{1}{16d}.$$

*Let* $\pi_1, \ldots, \pi_s$ *be sampled i.i.d.* $\sim (\hat{l}_1, \ldots, \hat{l}_n)$, *where* $\hat{l}_i = \mathbf{x}_i^\top \mathbf{A}^{-1} \mathbf{x}_i$. *If* $s \geq 8d^2$, *then*

$$\text{for} \quad \mathbf{Q}_\pi = \sum_{j=1}^{s} \frac{d}{\hat{l}_{\pi_j}} \mathbf{e}_{\pi_j} \mathbf{e}_{\pi_j}^\top, \qquad \frac{\det(\frac{1}{s}\mathbf{X}^\top \mathbf{Q}_\pi \mathbf{X})}{\det(\mathbf{A})} \leq 1 \quad \text{and} \quad \mathbb{E}\left[\frac{\det(\frac{1}{s}\mathbf{X}^\top \mathbf{Q}_\pi \mathbf{X})}{\det(\mathbf{A})}\right] \geq \frac{3}{4}.$$

Proof of Lemma 15 follows along the same lines as the proof of Theorem 6. We can compute matrix $\mathbf{A}^{-1}$ efficiently in time $\widetilde{O}(nd + d^3/\epsilon^2)$ using a sketching technique called Fast Johnson-Lindenstraus Transform [1], as described in [16]. However, the cost of computing the entire rescaling distribution is still $O(nd^2)$. Standard techniques circumvent this issue by performing a second matrix sketch. We cannot afford to do that while at the same time preserving the sufficient quality of leverage score estimates needed for leveraged volume sampling. Instead, we first compute weak estimates $\tilde{l}_i = (1 \pm \frac{1}{2})l_i$ in time $\widetilde{O}(nd + d^3)$ as in [16], then use rejection sampling to sample from the more accurate leverage score distribution, and finally compute the correct rescaling coefficients just for the obtained sample. Note that having produced matrix $\mathbf{A}^{-1}$, computing a single leverage score estimate $\hat{l}_i$ takes $O(d^2)$. The proposed algorithm with high probability only has to compute $O(s)$ such estimates, which introduces an additional cost of $O(sd^2) = O((k + d^2)\,d^2)$. Thus, as long as $k = O(d^3)$, dominant cost of the overall procedure still comes from the estimation of matrix $\mathbf{A}$, which takes $\widetilde{O}(nd + d^5)$ when $\epsilon$ is chosen as in Lemma 15.

It is worth noting that *fast leveraged volume sampling* is a valid $q$-rescaled volume sampling distribution (and not an approximation of one), so the least-squares estimators it produces are exactly unbiased. Moreover, proofs of Theorems 8 and 9 can be straightforwardly extended to the setting where $q$ is constructed from approximate leverage scores, so our loss bounds also hold in this case.