[Reviews · NeurIPS 2018]

Reviewer 1



The paper explores a connection between volume sampling and linear regression. Having an input matrix A and a response vector b, it wants to solve linear regression of (A,b) by sampling few rows of A, and yet get a good estimation to optimal least square linear regression weight vector w*. They proposed a leverage score based volume sampling method (called determinantal rejection sampling in the paper) that samples O(dlog d + d/eps) rows of the input matrix and gives a 1+eps multiplicative gaurantee on weight vector. First off, they showed a rescaled version of volume sampling that gives an unbiased estimate to w*. Afterwards, they specifically mention a volume sampling method that is using probabilities proportional to leverage scores. The paper is written clearly, the idea and the algorithms are novel and original. I did not verify all proofs, but the ones I checked seemed correct.

Reviewer 2



This paper studies deficiencies of volume sampling, and proposes a modification based on leverage scores, or renormalizing the current ellipse before performing volumne rejection sampling. It improves the number of unbiased samples required to guarantee 1\pm\epsilon accuracy by a factor of \epsilon^{-1}, and also demonstrates the good empirical performances of its routines on datasets from LibSVM (in Supplementary materials E). Both linear regression and volume sampling are well studied topics, and the observations made in this paper are quite surprising. The paper clearly outlines a class of matrices that are problematic for volume sampling, and then proves the properties of the revised methods. The proposed methods also exhibit significant empirical gains over other methods in the small sample size regime, which are arguable the more important cases. I believe these contributions are of significant interest to the study of both randomized sampling and randomized numerical linear algebra.

Reviewer 3



This paper continues a last year NIPS paper regarding volume sampling for the regression problem. It improved and somewhat completes the last result by showing the the original method may get sup-optimal results and then establishing a variant of the volume sampling so it’s will not suffer from this kind of pitfalls while maintaining the benefits of volume sampling (especially the unbiased). Along the way the paper uses a new “Determinantal rejection sampling” method and a few mathematical results which may be of independent interest. Overall the paper is well written